# FROM WORDS TO AMINO ACIDS: DOES THE CURSE OF DEPTH PERSIST?

**Aleena Siji**[1,2][*][‡]   **Amir Mohammad Karimi Mamaghan**[3][*][‡]   **Ferdinand Kapl**[1,2]   **Tobias Höppe**[1,2]
**Emmanouil Angelis**[1,2]   **Andrea Dittadi**[1,2]   **Maurice Brenner**[1,4]   **Michael Heinzinger**[1,4]
**Karl Henrik Johansson**[3]   **Kaitlin Maile**[5]   **Johannes von Oswald**[5]   **Stefan Bauer**[1,2]

[1] Technical University of Munich   [2] Helmholtz AI, Munich   [3] KTH Royal Institute of Technology
[4] Institute of Computational Biology, Helmholtz Munich   [5] Google, Paradigms of Intelligence Team

## ABSTRACT

Protein language models (PLMs) have become widely adopted as general-purpose models, demonstrating strong performance in protein engineering and de novo design. Like large language models (LLMs), they are typically trained as deep transformers with next-token or masked-token prediction objectives on massive sequence corpora and are scaled by increasing model depth. Recent work on autoregressive LLMs has identified the *Curse of Depth*: later layers contribute little to the final output predictions. These findings naturally raise the question of whether a similar depth inefficiency also appears in PLMs, where many widely used models are not autoregressive, and some are multimodal, accepting both protein sequence and structure as input.In this work, we present a depth analysis of six popular PLMs across model families and scales, spanning three training objectives, namely autoregressive, masked, and diffusion, and quantify how layer contributions evolve with depth using a unified set of probing- and perturbation-based measurements. Across all models, we observe consistent depth-dependent patterns that extend prior findings on LLMs: later layers depend less on earlier computations and mainly refine the final output distribution, and these effects are increasingly pronounced in deeper models. Taken together, our results suggest that PLMs exhibit a form of depth inefficiency, motivating future work on more depth-efficient architectures and training methods.

## 1 INTRODUCTION

Proteins are essential macromolecules in living organisms. They are specified by amino acid sequences, which largely determine a protein's three-dimensional structure, and in turn, its function. Recent progress in protein modeling has been driven by protein language models (PLMs), which are large-scale models trained on evolutionary-scale sequence databases using self-supervised objectives, and can capture meaningful biochemical and evolutionary properties even without explicit structural or functional information (Lin et al., 2023; Nijkamp et al., 2023; Ferruz et al., 2022; Wang et al., 2024). As a result, PLM representations can be used as general-purpose features for many downstream problems, including protein function annotation (Brandes et al., 2022; Rives et al., 2021; Elnaggar et al., 2021), structure-related prediction tasks such as secondary structure and contact prediction (Rao et al., 2019; Lin et al., 2023), and mutation effect prediction (Meier et al., 2021; Nijkamp et al., 2023). More recently, PLMs have rapidly moved beyond sequence-only encoders. Several recent models extend the PLMs by incorporating additional modalities, most commonly the structural information, and are trained with objectives that enable generation (Hayes et al., 2025; Wang et al., 2025; Geffner et al., 2025; Chen et al., 2025a). This makes them useful not only as feature extractors but also as generative models which can be utilized for protein engineering and design.

---

[*]Equal contribution.
[‡]Correspondence: `aleena.siji@helmholtz-munich.de, amkm@kth.se`.

This rapid progress has largely followed the same scaling and training paradigm that has shaped modern large language models (LLMs). In particular, PLMs and LLMs are typically built on the Transformer architecture (Vaswani et al., 2017) and trained with self-supervised language modeling objectives on massive sequence corpora: natural language text for LLMs and large protein sequence (and sometimes, structure) databases for PLMs (Rives et al., 2021; Lin et al., 2023). In both cases, the model learns contextual token representations by predicting tokens from context, either by masking random input tokens (MLM) or by predicting the next token with causal masking. As a result, many of the practical questions that arise when scaling LLMs also naturally apply to PLMs.

As an example of such scaling questions, recent analyses of modern autoregressive LLMs suggest that simply stacking more transformer blocks does not necessarily translate into proportional gains in model capability. Sun et al. (2025) formalize this as the *Curse of Depth*: across several popular autoregressive LLM families, deeper layers often contribute much less than earlier layers, and pruning or perturbing many late layers causes only a small performance change. Csordás et al. (2025) reach a consistent conclusion by directly analyzing the residual stream. They find a sharp drop in layer contributions around the middle of the network, and show that skipping layers in the second half has a much smaller effect on subsequent computations, suggesting that many late layers mainly refine the final output distribution rather than building reusable intermediate results. Together, these findings raise a broad concern for deep transformer models. In particular, increased depth may be used inefficiently, with a substantial fraction of layers being less effective during training and inference (Sun et al., 2025; Csordás et al., 2025).

These findings in LLMs naturally raise the question of whether a similar depth inefficiency also appears in PLMs, where many widely used models are not autoregressive, and some are multimodal, incorporating additional information such as protein structure. While PLMs closely mirror LLMs in both architecture and training, this issue has not been systematically analyzed for PLMs. Furthermore, because of the key differences in modality and common training objectives, existing results for LLMs do not necessarily transfer to PLMs. In this work, we present a comprehensive depth analysis of a diverse set of widely used PLMs across model families, sizes, and training objectives, and quantify how layer contributions and representation changes evolve with depth using a unified suite of probing- and perturbation-based measurements. In particular, our contributions are as follows:

1. We present the first systematic study of depth usage in protein language models. We analyze 6 widely used PLM families with 20 model variants in total, covering autoregressive, masked, and diffusion objectives, and including two multimodal models.

2. Building on recent depth analyses in LLMs (Csordás et al., 2025), we develop a unified depth-analysis framework for PLMs that combines probing- and perturbation-based measurements with layer-wise downstream evaluation on the popular ProteinGym benchmark (Notin et al., 2023). Using this framework, we find a depth-usage pattern consistent with observations in LLMs: across all models, intermediate layers contribute most to downstream performance as models scale, while later layers primarily provide incremental refinement of the final predictions

3. We show that this behavior generalizes beyond next-token prediction and also emerges under masked and diffusion objectives and persists in multimodal PLMs. This suggests that depth inefficiency is a general property of modern PLMs and motivates more depth-efficient architectures and training methods.

## 2 RELATED WORK

**Protein Language Models.** In recent years, alongside the rapid progress of LLMs, PLMs have also evolved quickly in terms of architecture, modality, and training objectives. Early large-scale sequence models showed that transformers trained on billions of amino acids learn transferable representations of protein grammar for diverse downstream tasks (Brandes et al., 2022; Elnaggar et al., 2021). Building on this, the ESM family scaled bidirectional transformer encoders from ESM-1b (Rives et al., 2021) to ESM-2 (Lin et al., 2023), improving single-sequence representations and enabling strong structure-related predictions. More recently, PLMs have expanded beyond sequence-only inputs: Hayes et al. (2025) propose ESM-3, trained over sequence, structure, and function, enabling conditional generation over any modality. In parallel, PLMs have diversified in training objectives. Autoregressive models such as the ProGen family (Madani et al., 2020; Nijkamp et al.,

2023) and ProtGPT2 (Ferruz et al., 2022) support controllable sequence generation, while Wang et al. (2024) introduce discrete diffusion for protein sequences and Wang et al. (2025) extend it with structure for joint sequence–structure modeling in protein design. Finally, retrieval-augmented approaches such as Profluent-E1 (Jain et al., 2025) leverage homolog retrieval to improve zero-shot predictions. Together, these lines of work reflect a broader shift toward multimodal, generative, and retrieval-enhanced PLMs for both prediction and design.

**Depth-wise Analysis of Transformers.** Depth analysis of Transformer-based models has evolved from probing what layers encode to intervening on layers to test their contribution to the output. Early works used probes and attention analyses to map linguistic structure across depth, revealing layer-wise differences in what is linearly recoverable from representations (Hewitt & Manning, 2019; Tenney et al., 2019; Clark et al., 2019). More recent intervention studies show that autoregressive language models can tolerate dropping or swapping many deeper layers, motivating stage-like views of inference (Lad et al., 2024). Complementing this, Skean et al. (2025) identify mid-layer bottlenecks and show that intermediate layers can outperform the final layer on many embedding-style tasks. Two recent works then consolidate these observations into a depth-inefficiency view: Csordás et al. (2025) find a mid-depth transition after which later layers have much smaller downstream influence, and Sun et al. (2025) formalize this as the *Curse of Depth* and propose Layer Norm Scaling to improve deeper-layer learning. Inspired by these findings, which are mostly established for autoregressive LLMs, we ask whether a similar depth inefficiency arises in PLMs. We study layer contributions across autoregressive, masked, and diffusion PLMs, and include multimodal models.

## 3 PROBLEM SETUP

In this section, we briefly describe the PLM families included in our study and the evaluation setting used to assess depth efficiency across models and scales.

**Models.** Our model set is chosen to reflect the main training paradigms and to cover both sequence-only and multimodal settings. In particular, the selected models vary along two main axes that are central to our study: i) they cover different training objectives, namely masked-token prediction, autoregressive next-token prediction, and discrete diffusion, and ii) they differ in input modality, ranging from sequence-only encoders to models that incorporate additional protein signals such as structural and functional information. As sequence-only PLMs, we include ESM2 (Lin et al., 2023), a widely-used family of masked encoders that serve as a standard baseline, DPLM (Wang et al., 2024), Profluent-E1 (Jain et al., 2025), and ProGen2 (Nijkamp et al., 2023). Notably, while E1 is sequence-only, it is trained in a retrieval-augmented setting by conditioning on additional unaligned homolog sequences. As multimodal PLMs, we include ESM3 (Hayes et al., 2025), a masked model trained over protein sequence, structure, and function, and DPLM2 (Wang et al., 2025), an upgraded version of DPLM trained jointly over protein sequence and structure. Furthermore, to enable consistent comparisons across all families, we restrict our analysis to the sequence input pathway for every model, including multimodal models. Finally, for every model family, we consider all publicly available sizes, which allows us to quantify how depth-dependent effects evolve with scale. A full description of the models is provided in Appendix A.1.

It is worth noting that some model families also differ in their pretraining datasets and data scales, which could, in principle, influence the observed behavior. Since our goal is not to benchmark or rank models against each other, we do not attempt to control for or attribute effects to these data differences. Instead, we focus on depth-dependent patterns *within* each model family by comparing different sizes within the same family, and treat cross-family comparisons as out of scope. Importantly, within a given model family, models are typically trained on the same pretraining dataset with broadly comparable training setups and often similar token budgets, which helps isolate depth-related trends.

**Evaluations.** Our primary analysis follows the probing- and intervention-based depth analysis of Csordás et al. (2025). We quantify how representations and output predictions evolve with depth, and measure layer contributions via controlled perturbations and their downstream effects on subsequent computations. Furthermore, to link these intrinsic measurements to practical utility, we also evaluate layer-wise downstream performance on ProteinGym (Notin et al., 2023), a standard benchmark for zero-shot mutation-effect prediction based on deep mutational scanning assays. Further details are provided in Section 4 and Appendix A.2.

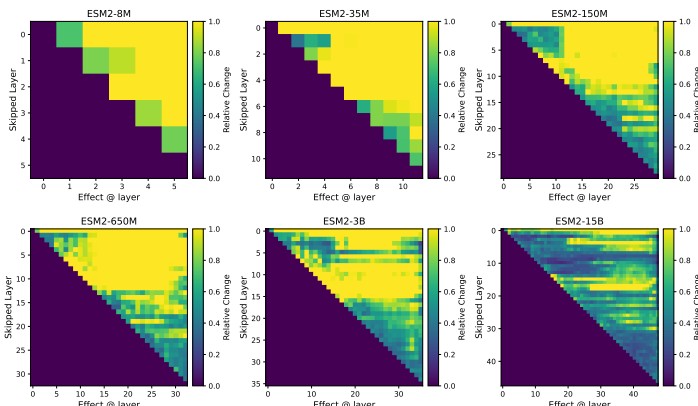

Figure 1: **Maximum propagated effect of skipping each layer on future-token computations in ESM2.** Even at 35M, skipping later layers produces relatively weak propagated effects compared to skipping early layers. From 150M onward, this separation becomes clear: a substantial fraction of late layers can be skipped with only minor changes in subsequent computations on future tokens. This pattern strengthens with scale, indicating that downstream sensitivity increasingly concentrates in early-to-middle layers, while later layers mainly refine the final prediction. We also observe localized low-effect regions among early layers, suggesting that not all early layers contribute equally. This aligns with the stage-wise view of Lad et al. (2024) and suggests that depth is organized into multiple inference stages with weaker dependencies across certain layer ranges.

**Limitations.** While our goal is to provide a robust and informative study of depth efficiency in PLMs, our analysis has limitations in model coverage and modality. First, we mainly focus on PLMs and do not include all-atom co-generation models, which have recently become more popular and have shown strong performance for joint sequence–structure generation and structure-conditioned design (Ingraham et al., 2023; Geffner et al., 2025; Chen et al., 2025a). Second, our multimodal coverage is limited to two models, and for consistency across sequence-only and multimodal models, we analyze only the sequence input pathway of multimodal PLMs, ignoring the structure stream. A broader study that includes more multimodal and all-atom models, and that contrasts depth-dependent behavior across different input streams, is a natural direction for future work.

## 4 EXPERIMENTS

We conduct a set of probing- and intervention-based experiments to analyze depth usage from different perspectives. First, we intervene on each layer and measure its contribution to downstream computations (Section 4.1). Next, we study how the model's prediction distribution evolves with depth by extracting a layer-wise implied token distribution and comparing it to the final-layer distribution (Section 4.2). Finally, in Section 4.3, we evaluate downstream utility via layer-wise performance on ProteinGym and analyze when strong predictive signals emerge across the network. Unless stated otherwise, experiments in Sections 4.1 and 4.2 use 40 protein sequences randomly sampled from UniRef50 (Suzek et al., 2007) as input prompts. Full experimental details are provided in Appendix A.2. We focus on ESM2 as a representative and widely used PLM in the main text, and report results for the other models in Appendix B.

### 4.1 HOW MUCH DOES EACH LAYER CONTRIBUTE TO LATER COMPUTATIONS?

We measure how much each layer contributes to *later* computations by skipping that layer for earlier token positions and tracking how the intervention propagates downstream. Concretely, we sample a split position and run a second forward pass where a chosen source layer is skipped only up to the split. We then quantify the resulting change in subsequent-layer representations on the remaining (future) token positions and report the propagated effects.

The original setup of Csordás et al. (2025) is tailored to autoregressive next-token prediction. For ProGen2, we therefore follow the same procedure and skip a layer only up to a randomly sampled token position and measure its effect on future tokens. For masked and diffusion-based models,

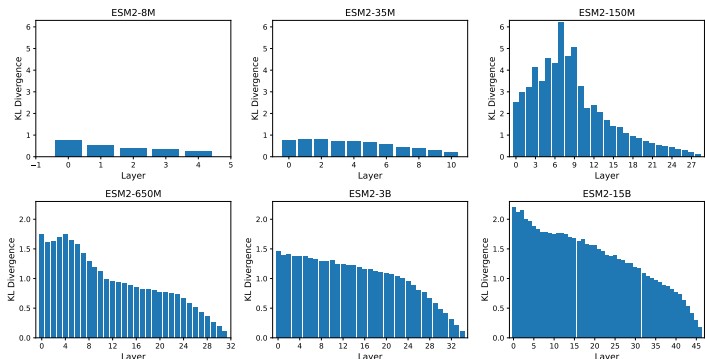

Figure 2: **KL divergence between the LogitLens layer-wise output distribution and the final output distribution for ESM2, plotted across depth.** From 150M onward, the KL divergence decreases steadily toward later layers, indicating that deeper layers make increasingly incremental updates that bring the distribution closer to the final prediction. For the largest variants, the KL divergence is already relatively low in earlier layers, making the late-layer refinement phase less sharply separated. Overall, the trend remains consistent with later layers primarily refining an increasingly stable prediction.

which are not trained with a causal left-to-right objective, we adapt this idea by masking 15% of tokens, intervening on a random subset of token positions, and measuring effects on the remaining non-intervened masked tokens. Specifically, we sample 20–80% of masked positions and 20–80% of non-masked positions for intervention, and compute effects on the remaining masked positions. We also measured effects on all non-intervened tokens (i.e., masked and non-masked), and observed similar trends. Hence, we report results only for the former.

**Results.** Figure 1 and Figures 5 to 9 in Appendix B.1 show the propagated effects for each model family. Across all models, we observe a consistent pattern: skipping earlier layers typically causes larger changes in later layers, whereas skipping later layers often has only a minor effect on the remaining computation. For ESM2 (Figure 1), this trend is visible already at 35M, becomes clear from 150M onwards, and strengthens with scale. At 15B, skipping a large fraction of layers in the second half of the model has little effect on subsequent layers.

DPLM (Figure 5) shows a similar pattern starting from the smallest size (150M), and it becomes clearer with scale. DPLM2 (Figure 6) behaves similarly across all sizes. Profluent-E1 (Figure 7) follows the same overall trend, but with a somewhat different structure in which depth regions most strongly influence later layers. For ESM3 (Figure 8), later layers again have weaker effects, but the transition is less sharp: skipping layers in the first half produces smaller changes within the first-half computation than in the other models, and the low-effect regime in late layers is present but less sharp overall.

We additionally quantify how skipping each layer changes the final model outputs by measuring the change in output norm before and after the intervention, restricted to future tokens. For ESM2 (Figure 10), we observe an overall downward trend with depth, with small fluctuations and minor differences across model sizes. ESM3 (Figures 8 and 14) shows the opposite pattern: early layers have the weakest effect on future outputs, while the most impactful layers cluster around mid-depth, which is consistent with its skiplayer patterns. For DPLM, DPLM2, and Profluent-E1, we find a more consistent decrease with depth, where deeper layers tend to induce smaller changes in future outputs (Figures 11 to 13). This effect is particularly pronounced in ProGen2, where output-norm differences decrease with depth across all sizes (Figure 15).

Finally, for all non-autoregressive models (i.e., ESM2, ESM3, DPLM, DPLM2, and Profluent-E1), we observe two low-effect regions: one in the early layers and one in the later layers. This is evident both in the skiplayer heatmaps and in the output-norm trends, and suggests that depth may be organized into stages with weaker dependencies across certain layer ranges. This is consistent with the findings of Lad et al. (2024), which show that for autoregressive LLMs, transformer computation can be decomposed into several stages of inference. Taken together, the heatmaps and the output-norm trends suggest that similar stage-like patterns may also arise in masked and diffusion PLMs. We leave further investigation to future work.

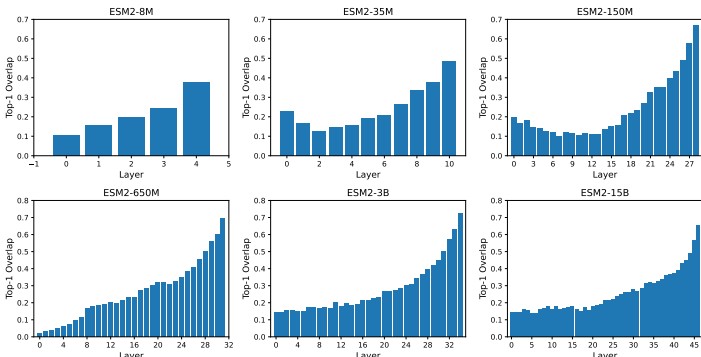

Figure 3: **Top-1 overlap between the layer-wise prediction and the full-model prediction for ESM2 across depth.** Across ESM2 variants, agreement is low in early layers and increases toward the end of the network. The increase is most pronounced in the final layers, especially for larger models, where it rises gradually through mid-depth and then more sharply near the final layers.

**Summary.** Overall, across all models and training objectives, we observe a consistent trend: as models scale, the propagated effects of skipping layers are dominated by early-to-middle layers, while skipping many late layers changes downstream computations only slightly. ESM3 is the only exception, where the weakest effects are concentrated in the earliest layers rather than the latest ones. This mirrors findings in LLMs and suggests a form of depth inefficiency, in which layers with weak propagated effects act mainly as incremental refinement steps on the final output rather than major drivers of subsequent computation (Lad et al., 2024; Sun et al., 2025; Csordás et al., 2025; Gromov et al., 2025).

## 4.2 How does the output distribution evolve across depth?

We further study how a model's output distribution evolves with depth using LogitLens (Nostalgebraist, 2020). Concretely, after each layer, we take the residual stream, apply the model's final normalization and unembedding to obtain token logits, and interpret the resulting softmax as the layer's implied prediction distribution. We then report the KL divergence between this layer-wise distribution and the final distribution produced by the full model. In addition, we measure how often the top-$k$ tokens under the layer-wise distribution overlap with the top-$k$ tokens under the final distribution. Since protein vocabularies are much smaller than natural-language vocabularies and are essentially limited to the standard amino-acid alphabet ($|\mathcal{V}| \approx 20$), we report top-$k$ overlap with $k = 1$. Furthermore, to adapt this to masked and diffusion-based models, we follow the same setup as in Section 4.1 and compute the results for the masked positions.

**Results.** The results are shown in Figure 2 and Figures 16 to 20 in Appendix B.2. The KL-divergence curves show a consistent trend across most model families: deeper layers produce layer-wise distributions that progressively approach the model's final distribution. For ESM2 (Figure 2), this trend is visible at smaller scales and becomes clear from 150M onward, where the KL divergence steadily decreases toward later layers; the same pattern holds for larger variants. DPLM also shows decreasing KL divergence with depth, indicating increasingly incremental changes to the final distribution in later layers (Figure 16). Profluent-E1 exhibits a similarly clear downward trend across all sizes (Figure 18). ESM3 follows the same overall behavior, with KL divergence steadily decreasing toward the end of the network (Figure 19).z

For the remaining models, the same picture largely holds, with a couple of nuances. DPLM2 shows a strong drop toward later layers, but for the largest size (3B), the trend is less clean and the KL divergence is already relatively low earlier in the network (Figure 17). ProGen2 shows the decreasing pattern for the large variant, whereas the xlarge variant has a much flatter profile with low KL divergence across most layers (Figure 20). Together, these cases indicate that for the largest variants, the layer-wise distributions may approach the full-model output earlier in depth. This could imply that later layers spend more of their capacity on incremental refinement, which would be consistent with stronger depth inefficiency at scale.

Furthermore, the top-1 overlap results in Figure 3 and Figures 21 to 25 in Appendix B.2 provide a complementary view of this refinement behavior. For ESM2 (Figure 3), agreement with the final

prediction is generally low in early layers and increases toward the end of the network. The increase is most pronounced in larger variants, where overlap rises gradually through mid-depth and then grows more sharply in the final layers. Smaller models show the same trend but more weakly, suggesting that intermediate layers still shape the representation while later layers increasingly align the prediction with the full-model output. We observe a similar behavior in the other model families and refer to Appendix B.2 for full results.

**Summary.** Overall, the results are consistent with prior depth analyses in LLMs (Sun et al., 2025; Csordás et al., 2025; Skean et al., 2025; Gromov et al., 2025) and align with our observations in Section 4.1. For almost all model families, later layers tend to make smaller, incremental updates that mainly refine an increasingly stable output distribution, rather than inducing large shifts in the model's predictions, and this refinement behavior becomes clearer with scale in many cases.

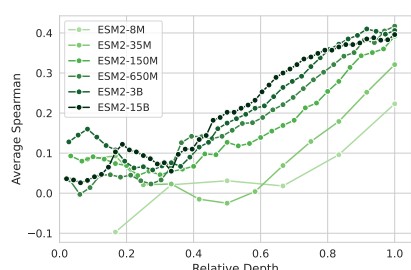

Figure 4: **Layer-wise ProteinGym performance for ESM2.** Average Spearman correlation as a function of relative depth, normalized to $[0, 1]$, where predictions are taken from each layer. Performance improves with depth for all model sizes, but the largest models exhibit diminishing returns in the final layers, suggesting that earlier layers already capture much of the signal and later layers mainly provide small refinements to final predictions.

## 4.3 HOW DOES DOWNSTREAM PERFORMANCE VARY ACROSS DEPTH?

To further analyze the contributions of different layers, we connect the intrinsic measurements in Sections 4.1 and 4.2 to practical utility by evaluating layer-wise performance on ProteinGym (Notin et al., 2023). ProteinGym is a large-scale benchmark for mutation effect prediction that contains 217 standardized deep mutational scanning (DMS) assays spanning millions of mutated sequences across diverse proteins and phenotypes. Following the official ProteinGym evaluation framework, we score variants in a zero-shot manner, compute Spearman's rank correlation between predicted scores and experimental measurements for each assay, and report the average Spearman correlation across all assays.

We focus on the four sequence-only models in our study: DPLM, ESM2, ProGen2, and Profluent-E1, using the official ProteinGym codebase. For each layer, we compute mutation scores from that layer's logits using the model's standard prediction head, without additional fine-tuning. For ESM2 and Profluent-E1, we use the masked-marginal scoring procedure of Meier et al. (2021). For DPLM, we apply the same masked-marginal protocol. For the autoregressive ProGen2, we use the likelihood-ratio scoring provided by the official ProteinGym implementation.

**Results.** Figure 4 and Figures 26 to 28 (with full phenotype breakdowns in Figures 29 to 32) in Appendix B.3 summarize the downstream results. Across all models, performance generally improves with depth, but for sufficiently large models, most gains happen in early-to-middle layers, after which additional layers yield noticeably smaller improvements. The trend is also not always strictly monotonic: for some smaller variants and occasionally even in early depth for larger ones, intermediate layers can temporarily underperform shallower layers before performance recovers, suggesting a short adjustment phase before later-layer refinements.

In particular, for ESM2 (Figure 4), a clear "knee" emerges around the 650M scale: smaller variants continue to improve fairly steadily across depth, whereas larger models, i.e., 650M and above, reach strong performance by middle layers and then show diminishing gains towards the final layers. Furthermore, across all sizes, the first half of the layers does not improve the downstream performance, which further supports the existence of several stages of inference (Lad et al., 2024) in PLMs. DPLM (Figure 26) exhibits a similar pattern with earlier onset: even at 150M, we observe a mild saturation which becomes clearer as the model size increases. For Profluent-E1 (Figure 27), the saturation regime is present across all sizes, with most gains achieved in the last ∼40% of the layers. Finally, for ProGen2 (Figure 28), the plateau becomes clear from ProGen2-medium onward and is especially pronounced for large and xlarge variants, where performance changes little after roughly the first half of the network.

**Summary.** Overall, for large-enough PLMs, intermediate layers already capture most of the signal needed for strong downstream performance, while later layers provide smaller and incremental gains. This complements our probing and intervention analyses and further supports our main conclusion: across diverse PLM families and training objectives, model scaling leads to a depth inefficiency where most of the downstream-relevant computation is formed by intermediate layers and later layers mostly provide incremental refinement of the final prediction.

## 5 DISCUSSION

**Depth inefficiency extends beyond language and next-token prediction.** Our findings show that depth inefficiency is not limited to next-token prediction or natural language models. We observe a broadly similar two-phase behavior, i.e., early-to-mid layers contributing most of the computation and later layers primarily refining the output distribution, across masked and diffusion-based models as well as multimodal PLMs. This consistency across objectives and modalities suggests that the phenomenon is not task-specific, and may reflect a broader architectural property of deep Transformers. Prior work in LLMs has identified similar trends: Csordás et al. (2025); Lad et al. (2024) describe a stage-like structure of inference where later layers perform less compositional work and instead sharpen the final predictions independently. Our results in the protein domain are consistent with these observations and extend them into a biologically grounded setting, where representations are not just semantic but biochemical. These parallels raise the possibility that depth inefficiency may be a consequence of how residual stream updates accumulate in Transformers.

**Bridging adaptive computation ideas to the protein domain is still challenging.** Efforts to address this limitation in LLMs often rely on making computation more adaptive or compositional. For example, through Chain-of-Thought prompting (Wei et al., 2022) or depth-dynamic models like the Inner Thinking Transformer (Chen et al., 2025b). However, applying these ideas to protein models is not necessarily trivial. Protein sequences lack a clear notion of intermediate "reasoning steps", and their functional behavior emerges from distributed, nonlinear evolutionary constraints rather than symbolic logic. That said, certain ideas may still be applicable. Promising directions include staging generation such as coarse-to-fine structural scaffolding, or utilizing existing inductive biases (Jumper et al., 2021; Dauparas et al., 2022; Lin et al., 2023; Morehead & Cheng, 2024) or designing new ones that encourage reusable computation throughout the network. Developing such frameworks may help protein models move beyond the fixed-depth inefficiency and better match the compositional complexity of their domain.

## 6 CONCLUSION

In this work, we ask whether protein language models exhibit a depth inefficiency analogous to the "curse of depth" reported for autoregressive LLMs. We study this question across a broad set of popular PLM families, sizes, and training objectives, including autoregressive, masked, and diffusion models as well as two multimodal sequence–structure models. Using a unified suite of layer-wise measurements, we connect intrinsic signals of layer contribution to downstream utility via (i) layer-skipping interventions to quantify how much each layer affects downstream computations, (ii) layer-wise output readouts that map intermediate residual outputs through the model's final normalization and output projection to track how close the implied token distribution is to the full-model prediction across depth, and (iii) layer-wise downstream evaluation on the ProteinGym benchmark to measure how performance changes when predictions are taken from intermediate depth. Across these complementary views, we find a consistent picture: as models scale, downstream sensitivity and performance gains become increasingly concentrated in early-to-middle layers, while a growing fraction of later layers has limited propagated influence and mainly performs small, incremental refinements of an already strong prediction. While the exact transition depth and sharpness vary by model family and objective, the overall pattern is robust across mechanistic probes and downstream evaluation, supporting the conclusion that depth inefficiency is also a common feature of modern PLMs.

These findings suggest several directions for future work. First, recent growing approaches, where models are progressively expanded during pretraining, have been reported to improve depth efficiency, making it a natural next step to test whether such training schemes mitigate the depth ineffi-

ciency we observe in PLMs and to understand how they change layer contributions. Second, while our analysis focuses primarily on sequence-only PLMs and the sequence stream for the included multimodal models, an interesting extension is to analyze other modality streams, most notably structure, within multimodal PLMs to see whether depth is used differently across modalities and whether refinement and saturation emerge at different depths. As an additional direction, these insights could be turned into practical adaptive-compute methods for protein models, such as early-exit or layer-skipping policies guided by our measurements, to reduce inference cost while preserving downstream performance.

## MEANINGFULNESS STATEMENT

We consider a "meaningful representation of life" as one that captures biologically grounded structure, such as constraints from evolution, biophysics, and function, in a way that supports generalization to unseen proteins and enables reliable prediction and design. Meaningful representations should be stable, reusable across tasks, and reflect mechanistic signals rather than dataset-specific shortcuts. Our work contributes by analyzing where and how such representations emerge across depth in modern protein language models. By identifying systematic depth inefficiencies and when predictions stabilize, we provide guidance for building more depth-efficient models that allocate capacity to the layers where biologically relevant computation is formed.

## ACKNOWLEDGEMENTS

This work was partially supported by the Wallenberg AI, Autonomous Systems and Software Program (WASP), funded by the Knut and Alice Wallenberg Foundation, and by the Helmholtz Foundation Model Initiative, supported by the Helmholtz Association.

The computations were enabled by the Berzelius resource, provided by the Knut and Alice Wallenberg Foundation at the National Supercomputer Centre, and by the Gauss Centre for Supercomputing e.V. (www.gauss-centre.eu), which provided the required computing time through the John von Neumann Institute for Computing (NIC) on the GCS Supercomputer JUPITER — JUWELS (Jülich Supercomputing Centre, 2021) at Jülich Supercomputing Centre (JSC).

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

## A  EXPERIMENTAL DETAILS

In this section, we explain the included PLMs and provide additional details on the experiments used in our study.

### A.1  MODELS

**ESM2.** ESM2 (Lin et al., 2023) is a Transformer encoder protein language model trained with a masked language modeling (MLM) objective. It is commonly used as a general-purpose protein representation model across many different tasks, and for zero-shot fitness prediction. ESM2 is released in multiple sizes, from 8M to 15B, and is trained on UniRef50 (Suzek et al., 2007).

**ESM3.** ESM3 (Hayes et al., 2025) is a large-scale multimodal protein foundation model designed to jointly model protein *sequence*, *structure*, and *function*. It is trained with an MLM objective and is primarily used for multimodal generation and design-style tasks, where one can condition on partial information of any of the modalities and generate consistent outputs. The ESM3 family is released at multiple scales of 1.4B, 7B, and 98B parameters, but only the smaller version is publicly available and therefore, used in our study.

**Profluent-E1.** Profluent-E1 (Jain et al., 2025) is a family of protein Transformer encoder models trained with an MLM-style objective and designed for strong protein sequence representation learning. A key feature of E1 is that it can incorporate retrieved evolutionary context: homologous sequences are provided as additional inputs, and the model integrates this multi-sequence context through a dedicated attention scheme. E1 is released in three sizes, including 150M, 300M, and 600M parameters, and is trained for a large token budget on Profluent's Protein Atlas (Bhatnagar et al., 2025) and UniRef (Suzek et al., 2007).

**DPLM.** DPLM (Wang et al., 2024) is a diffusion-based protein language model trained under a discrete denoising diffusion objective. Rather than predicting masked tokens in one shot (MLM) or generating left-to-right (AR), DPLM learns to iteratively denoise corrupted sequences across diffusion timesteps while using bidirectional context. This makes DPLM useful both as a generative model via iterative sampling and as a representation learner that can be fine-tuned or used zero-shot for downstream prediction tasks. DPLM is released in multiple sizes, including 150M, 650M, and 3B, and is pretrained on UniRef50 (Suzek et al., 2007).

**DPLM2.** DPLM-2 (Wang et al., 2025) extends DPLM to a multimodal diffusion model that jointly models protein *sequence and 3D structure*. It converts 3D coordinates into discrete structure tokens and trains a unified denoising model over the combined token space, enabling tasks such as folding, inverse folding, and multimodal scaffolding through conditional diffusion. It is released in the same sizes as DPLM and is trained on a structure-supervised dataset built from experimental and high-quality synthetic structures. During training, the model is initialized from a pretrained DPLM and is further trained on the multimodal data.

**ProGen2.** ProGen2 (Nijkamp et al., 2023) is an autoregressive Transformer decoder model trained with next-token prediction. It is mainly used for protein sequence generation and for zero-shot fitness prediction. ProGen2 is released in multiple sizes, including small, medium, base, large, and xlarge models that scale up to 6.4B parameters, and is trained on a curated large-scale protein sequence corpora. Given that ProGen2-base has the same number of parameters as the medium variant, we exclude it from our study.

## A.2 EVALUATION SETUP

### A.2.1 PROPAGATED LAYER SKIPPING

We follow the layer-skipping tools of Csordás et al. (2025) to measure how much each layer contributes to downstream computation by removing that layer's residual update and tracking how the change propagates through later layers. For a Transformer with residual stream states $h_\ell$ (after layer $\ell$), we view each layer as applying an additive update $\Delta h_\ell = h_{\ell+1} - h_\ell$. Skipping a source layer $s$ is implemented by removing its update, i.e., setting $\bar{h}_{s+1} := \bar{h}_s$, and then continuing the forward pass normally to obtain intervened activations $\{\bar{h}_\ell\}_{\ell>s}$. We then quantify the propagated effect of skipping $s$ on a later layer $\ell$ by comparing the downstream representations under the original and intervened forward passes. In practice, we aggregate these changes across token positions and prompts using a simple summary statistic of calculating the average of maximum effects over relevant token positions over prompts, to produce the layer-by-layer heatmaps.

To isolate importance for future predictions, we use the *skiplayer future* variant. For autoregressive models (ProGen2), we sample a split position $1 < t_s < T - 1$, apply the layer-skip intervention only to token positions $t \leq t_s$, and then measure the effect only on positions $t > t_s$. This directly tests whether computations performed for earlier tokens are reused when predicting later tokens. For masked and diffusion-based models that do not have a left-to-right notion of "future", we adapt *skiplayer future* by replacing future tokens with held-out, non-intervened positions: we randomly sample an intervention set of positions and apply the skip only on that subset, and then measure propagated effects on the remaining non-intervened positions. Concretely, we sample 20–80% of

masked positions and 20–80% of non-masked positions for intervention, and compute effects on the remaining masked positions. We additionally measure effects on all non-intervened tokens and observe that both choices yield similar trends. Hence, we report results only for the former, as predictions for masked token positions are most sensitive to perturbations.

In addition to representation-level effects, we also quantify how skipping each layer changes the model's final outputs in the *skiplayer future* setting. For each skipped layer, we compute the $\ell_2$ change in the output vector between the original and intervened runs, $\|y - \bar{y}\|_2$, and report the maximum change over the evaluated future positions. We use this norm-based metric rather than KL divergence because it yields clearer, more stable visualizations for this intervention setting. For all experiments, we use 40 protein sequences randomly sampled from UniRef50 (Suzek et al., 2007) as input prompts.

### A.2.2   LAYER-WISE OUTPUT READOUTS WITH LOGITLENS

To track how the model's implied prediction distribution evolves across depth, we use LogitLens readout (Nostalgebraist, 2020) applied to intermediate residual representations. After each layer $\ell$, we take the residual stream at that depth and map it directly to token logits using the model's standard output readout, i.e., the final normalization followed by the output projection. Applying softmax gives a layer-wise distribution $p_\ell(\cdot)$ over the vocabulary. We compare this intermediate distribution to the final distribution $p_L(\cdot)$ produced by the full model at the last layer $L$.

We report two complementary summary metrics. First, we compute the KL divergence $D_{\mathrm{KL}}(p_L \,\|\, p_\ell)$ to measure how close the layer-wise distribution is to the final prediction. Second, we measure discrete agreement with the final prediction using top-$k$ overlap. Since protein vocabularies are much smaller than natural-language vocabularies, essentially limited to the amino-acid alphabet, we report top-1 overlap ($k = 1$): whether the most likely token under $p_\ell$ matches the most likely token under $p_L$. We aggregate these metrics over token positions and prompts on the same UniRef50 prompt set as in the previous experiments and report the results.

### A.2.3   LAYER-WISE PROTEINGYM EVALUATION

To connect previous depth measurements to practical utility, we evaluate layer-wise performance on ProteinGym (Notin et al., 2023), a large benchmark of mutation effect prediction with 217 standardized deep mutational scanning (DMS) assays. For each assay, the goal is to assign a score to each variant sequence and report Spearman's rank correlation between predicted scores and experimental measurements. We follow the official ProteinGym evaluation protocol and codebase, and focus on the four sequence-only model families used for this analysis: ESM2 (masked LM), DPLM (discrete diffusion, used in a masked-LM scoring mode at inference), ProGen2 (autoregressive), and Profluent-E1 (sequence-only model in our evaluation setup).

For each model, we compute a mutant–wild-type score using likelihood-based scoring as implemented in the official repository, with a layer-wise early-exit variant for our depth analysis. Concretely, for a chosen layer $\ell$, we obtain logits from that layer via early exit and compute mutation scores exactly as in the full model, but using the layer-$\ell$ logits. For masked- and diffusion-based models such as ESM2, DPLM, and Profluent-E1, we use masked-marginals scoring (Meier et al., 2021): for each mutated position, we mask that position, compute the log-probability of the mutant amino acid and the wild-type amino acid at that position, and take their difference. For multi-mutation variants, we take the sum of per-mutation differences. For ProGen2, which is an autoregressive model, we use the ProteinGym repository's autoregressive scoring procedure. We then report Spearman correlation per assay and summarize performance by averaging across all assays.

To make results comparable across model sizes with different numbers of layers, we plot performance against relative depth, normalizing layer index to $[0, 1]$ within each model. We also follow ProteinGym's standard handling of long sequences when needed so that scores are computed consistently under each model's maximum context length. Overall, this evaluation provides a downstream counterpart to the mechanistic measurements: it measures how quickly a useful predictive signal emerges with depth and where performance gains saturate when increasing the number of layers.

# B  ADDITIONAL RESULTS

In Section 4, we have shown depth analysis results primarily for ESM2. Here we provide the results for the remaining models.

## B.1  HOW DOES EACH LAYER AFFECT ITS CONSECUTIVE LAYERS?

Figures 5 to 9 report propagated *skiplayer future* effects for ESM3, Profluent-E1, DPLM, DPLM2, and ProGen2. Recall that *skiplayer future* isolates whether computations performed for earlier tokens are reused when predicting held-out (non-intervened) positions. We quantify propagation at the representation level as the maximum change in downstream-layer hidden states on the future token positions when skipping each source layer.

Figures 11 to 15 report the corresponding output-level sensitivity under the same setup, measured as the maximum $\ell_2$ change in the model outputs ($\|y - \bar{y}\|_2$). While representation-level propagation captures how strongly information flows forward through the network, output-level sensitivity reflects how much this propagated change ultimately affects the model's predictions. Across figures, sharp peaks indicate layers whose computations are strongly reused for future predictions, whereas flatter profiles suggest weaker reuse.

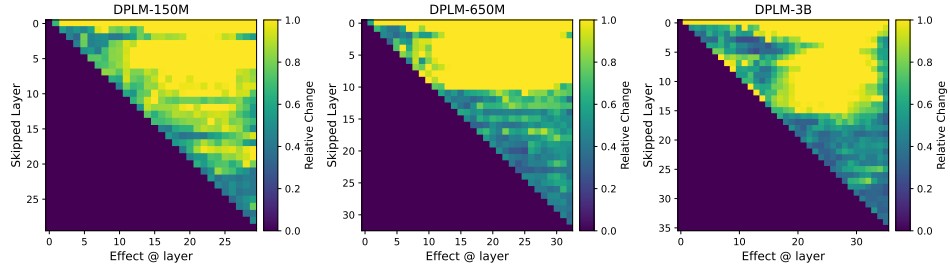

Figure 5: Maximum propagated effect of skipping each layer on future-token computations for DPLM.

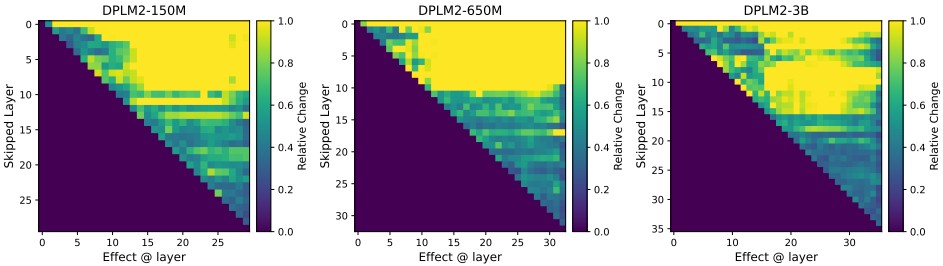

Figure 6: Maximum propagated effect of skipping each layer on future-token computations for DPLM2.

## B.2  HOW DOES THE PROBABILITY DISTRIBUTION VARY ACROSS LAYERS?

Figures 16 to 20 report how the model's token-level output distribution evolves across depth, using a layer-wise readout and comparing each layer's distribution to the final-layer distribution. Lower KL divergence indicates that a layer already produces a distribution close to the final model output, whereas higher KL divergence suggests that substantial refinement still occurs in later layers.

Complementing this distribution-level view, Figures 3, 21 to 23 and 25 report a decision-level metric: the top-1 overlap between the layer-wise argmax prediction and the final model's argmax prediction across depth. Top-1 overlap is easier to interpret but can hide meaningful distributional changes when probability mass shifts without changing the argmax, so we report both metrics together. Overall, these plots help distinguish whether later layers mainly sharpen already-formed predictions,

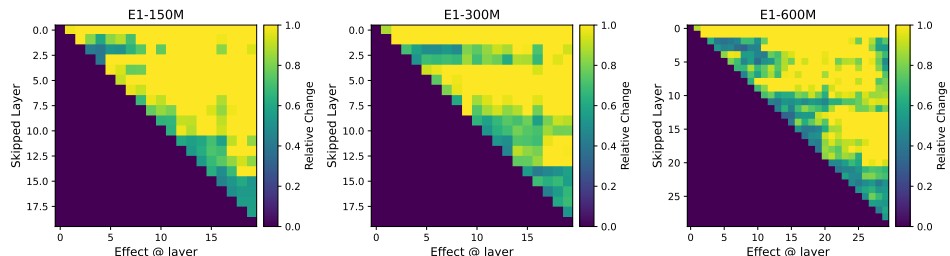

Figure 7: Maximum propagated effect of skipping each layer on future-token computations for Profluent-E1.

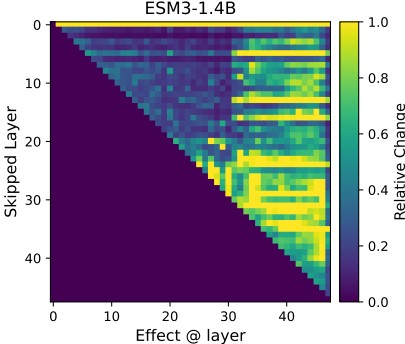

Figure 8: Maximum propagated effect of skipping each layer on later-layer representations (future tokens only), for ESM3.

i.e., low KL and high top-1 early, or whether they continue to meaningfully change the predicted distribution, by showing persistently high KL and low top-1.

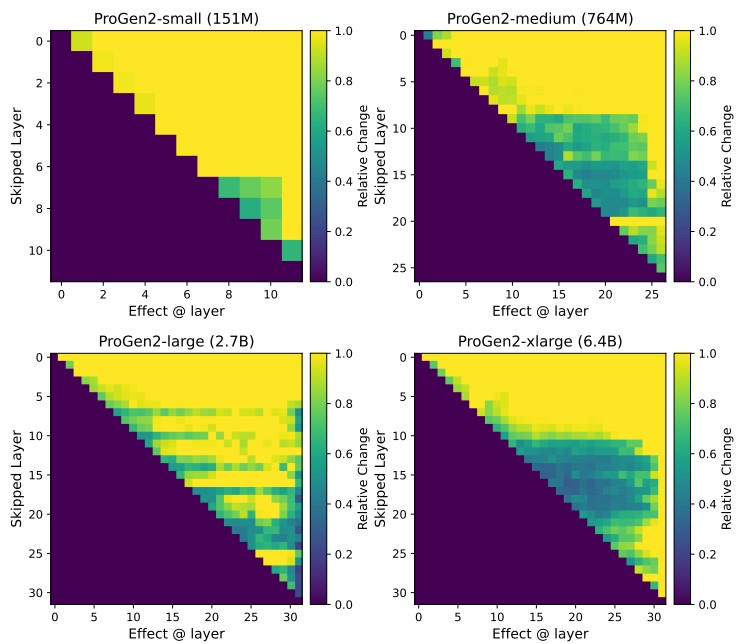

Figure 9: Maximum propagated effect of skipping each layer on future-token computations for ProGen2.

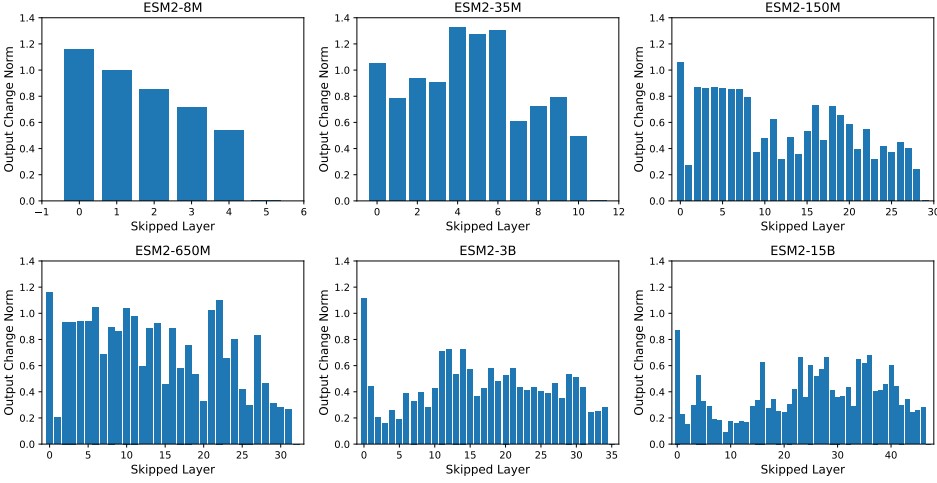

Figure 10: Maximum change in ESM2 output probabilities under layer skipping, restricted to future tokens only.

## B.3 HOW DO DIFFERENT LAYERS AFFECT DOWNSTREAM PERFORMANCE?

Figures 26 to 28 report layer-wise ProteinGym performance for DPLM, Profluent-E1, and ProGen2, measured as average Spearman correlation as a function of relative depth (normalized to $[0, 1]$). These curves characterize where useful information for zero-shot fitness prediction is most strongly expressed in the representation stack: rising performance with depth suggests that later layers refine task-relevant signals, whereas early plateaus indicate that additional depth provides limited marginal benefit for the downstream performance.

To assess whether these trends are consistent across task types, we additionally report the same analysis broken down by phenotype in Figures 29 to 32. The phenotype-level plots highlight which categories follow the overall average trend and which deviate by exhibiting earlier peaks or stronger

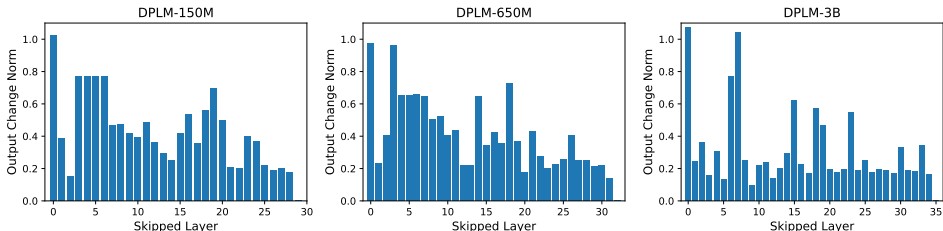

Figure 11: Maximum change in DPLM output probabilities under layer skipping, restricted to future tokens only.

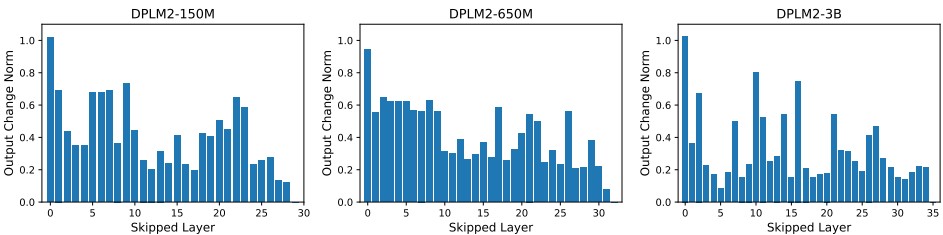

Figure 12: Maximum change in DPLM2 output probabilities under layer skipping, restricted to future tokens only.

reliance on deeper layers, providing a more fine-grained view of how depth affects downstream performance across different biological readouts.

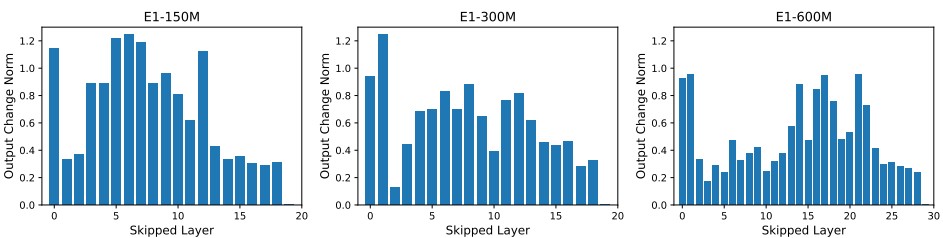

Figure 13: Maximum change in Profluent-E1 output probabilities under layer skipping, restricted to future tokens only.

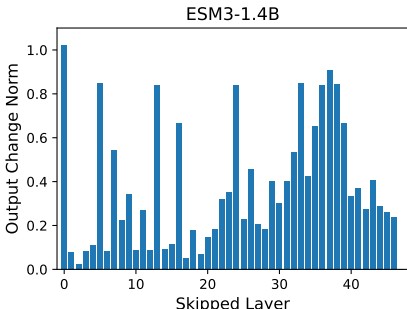

Figure 14: Maximum change in ESM3 output probabilities under layer skipping, restricted to future tokens only.

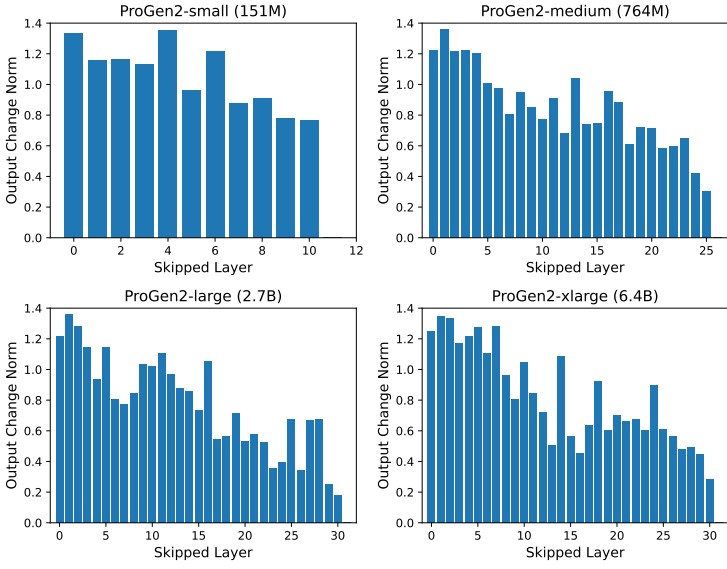

Figure 15: Maximum change in ProGen2 output probabilities under layer skipping, restricted to future tokens only.

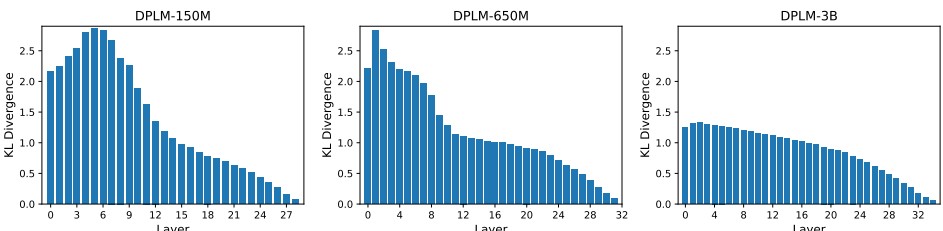

Figure 16: KL divergence between the LogitLens layer-wise output distribution and the final output distribution for DPLM, plotted across depth.

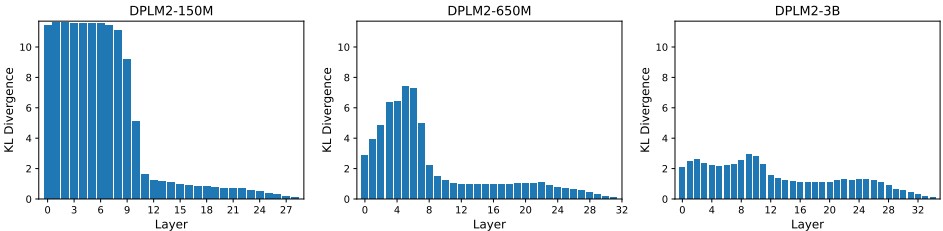

Figure 17: KL divergence between the LogitLens layer-wise output distribution and the final output distribution for DPLM2, plotted across depth.

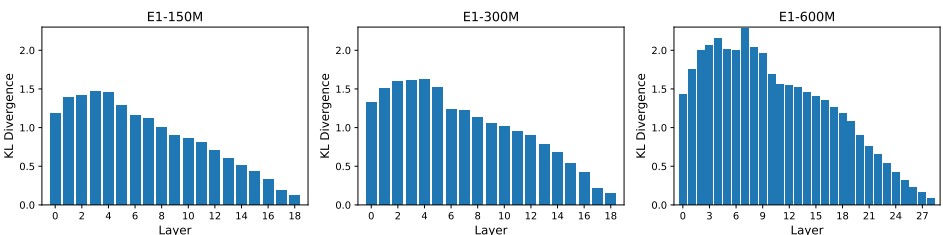

Figure 18: KL divergence between the LogitLens layer-wise output distribution and the final output distribution for Profluent-E1, plotted across depth.

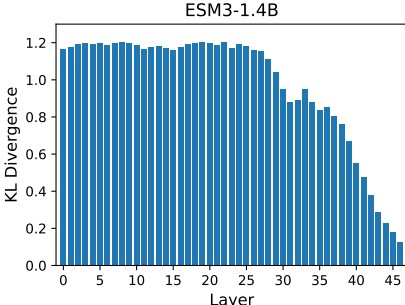

Figure 19: KL divergence between the LogitLens layer-wise output distribution and the final output distribution for ESM3, plotted across depth.

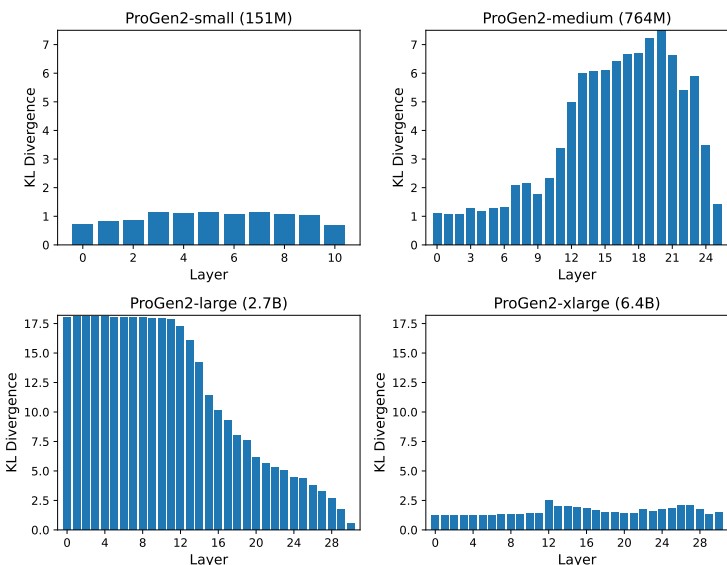

Figure 20: KL divergence between the output distribution from each layer and the model's final output distribution for ProGen2, plotted across depth.

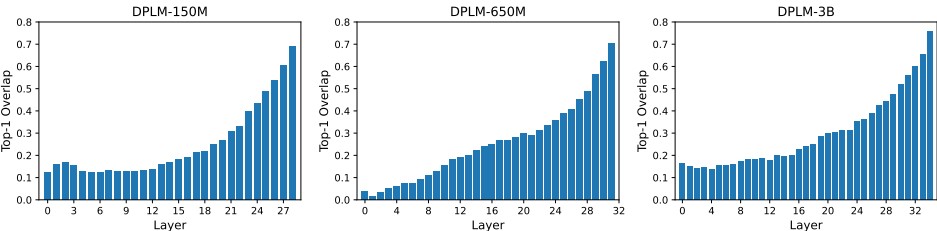

Figure 21: Top-1 overlap between the layer-wise prediction and the full-model prediction for DPLM across depth.

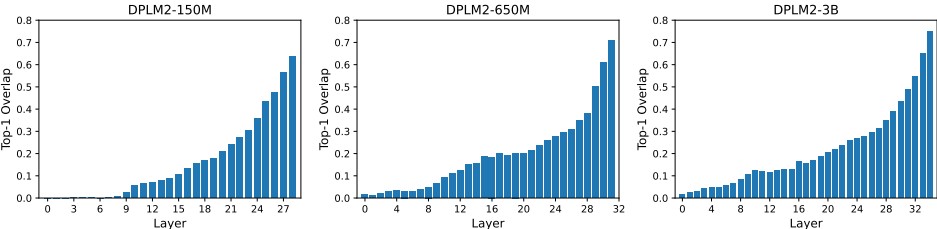

Figure 22: Top-1 overlap between the layer-wise prediction and the full-model prediction for DPLM2 across depth.

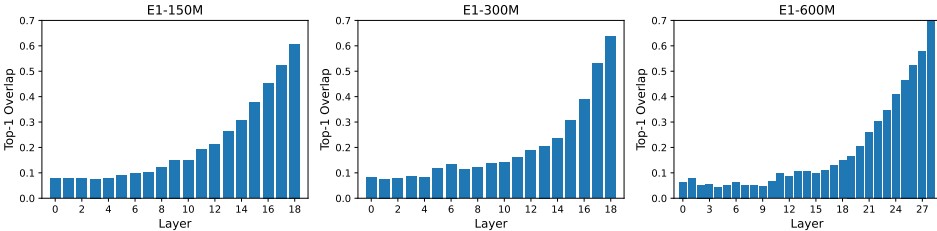

Figure 23: Top-1 overlap between the layer-wise prediction and the full-model prediction for Profluent-E1 across depth.

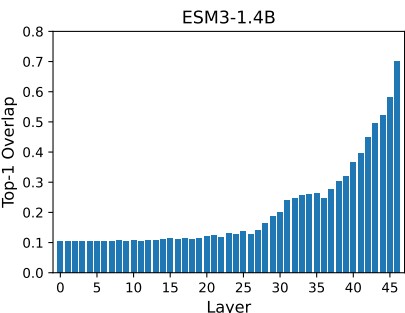

Figure 24: Top-1 overlap between the layer-wise prediction and the full-model prediction for ESM3 across depth.

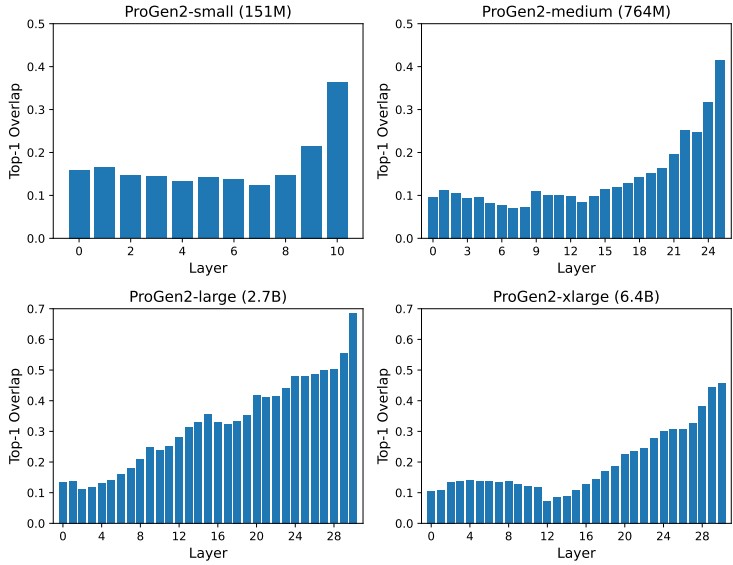

Figure 25: Top-1 overlap between the layer-wise prediction and the full-model prediction for ProGen2 across depth.

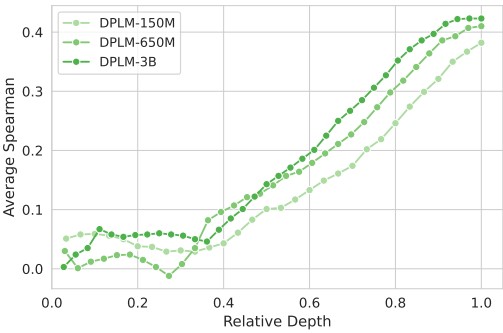

Figure 26: Average Spearman correlation for DPLM on ProteinGym, calculated at each layer. The relative depth is normalized to $[0, 1]$.

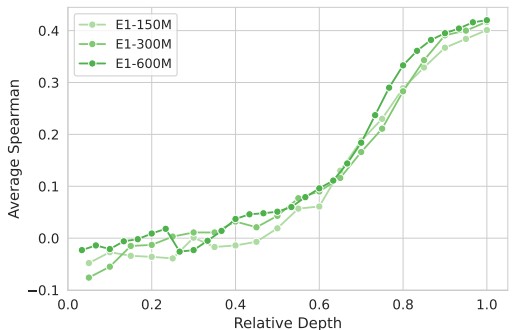

Figure 27: Average Spearman correlation for Profluent-E1 on ProteinGym, calculated at each layer. The relative depth is normalized to $[0, 1]$.

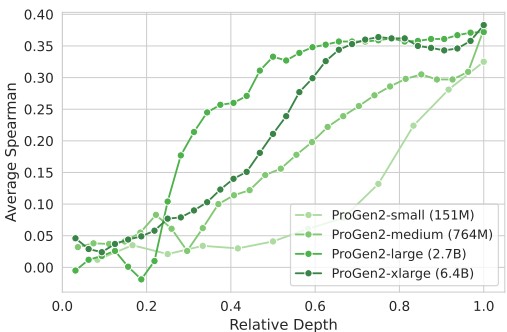

Figure 28: Average Spearman correlation for ProGen2 on ProteinGym, calculated at each layer. The relative depth is normalized to $[0, 1]$.

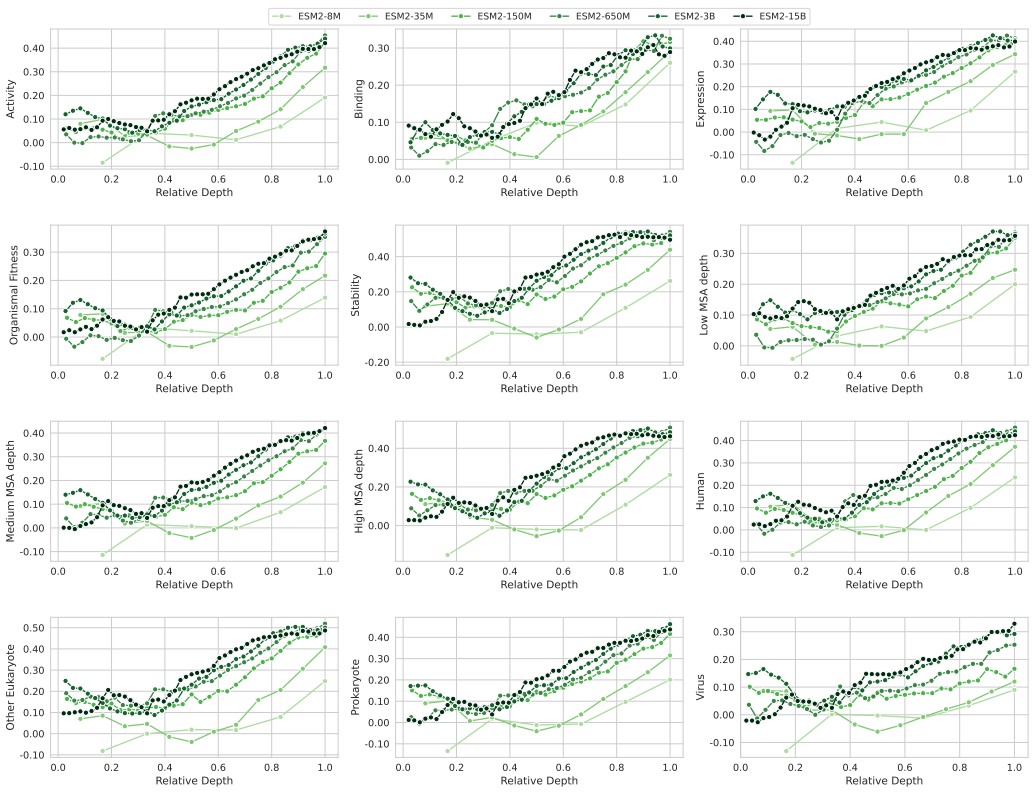

Figure 29: Average Spearman correlation for ESM2 on ProteinGym, computed at each layer and shown separately by phenotype. Relative depth is normalized to $[0, 1]$.

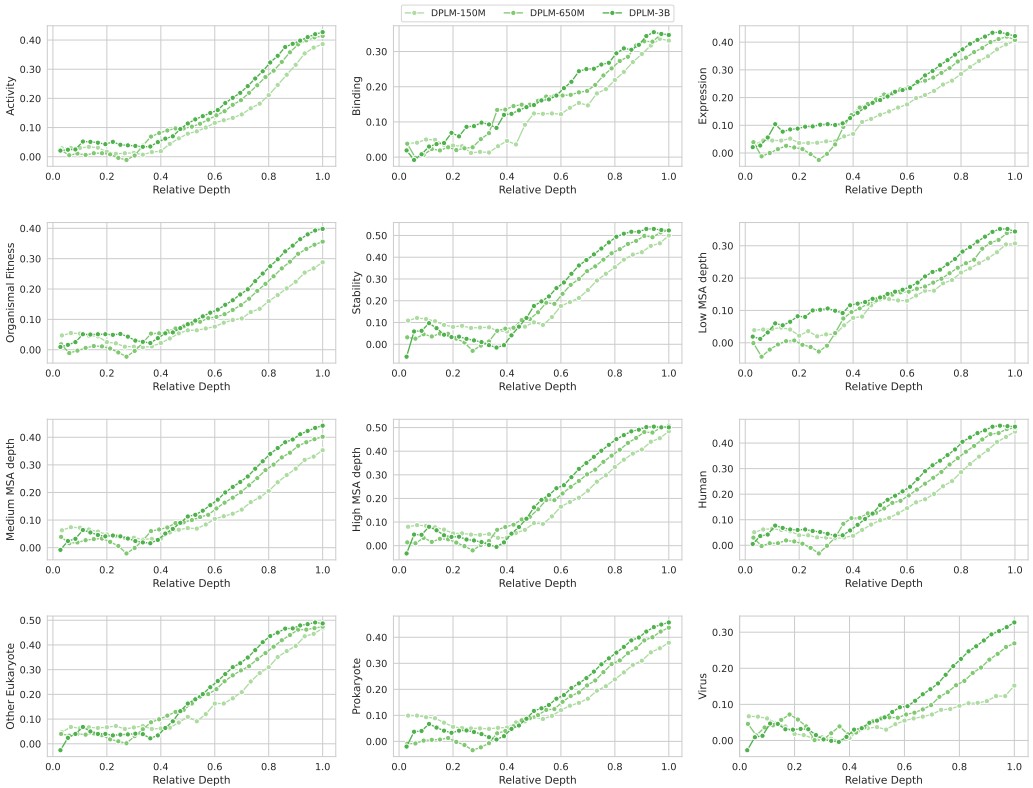

Figure 30: Average Spearman correlation for DPLM on ProteinGym, computed at each layer and shown separately by phenotype. Relative depth is normalized to $[0, 1]$.

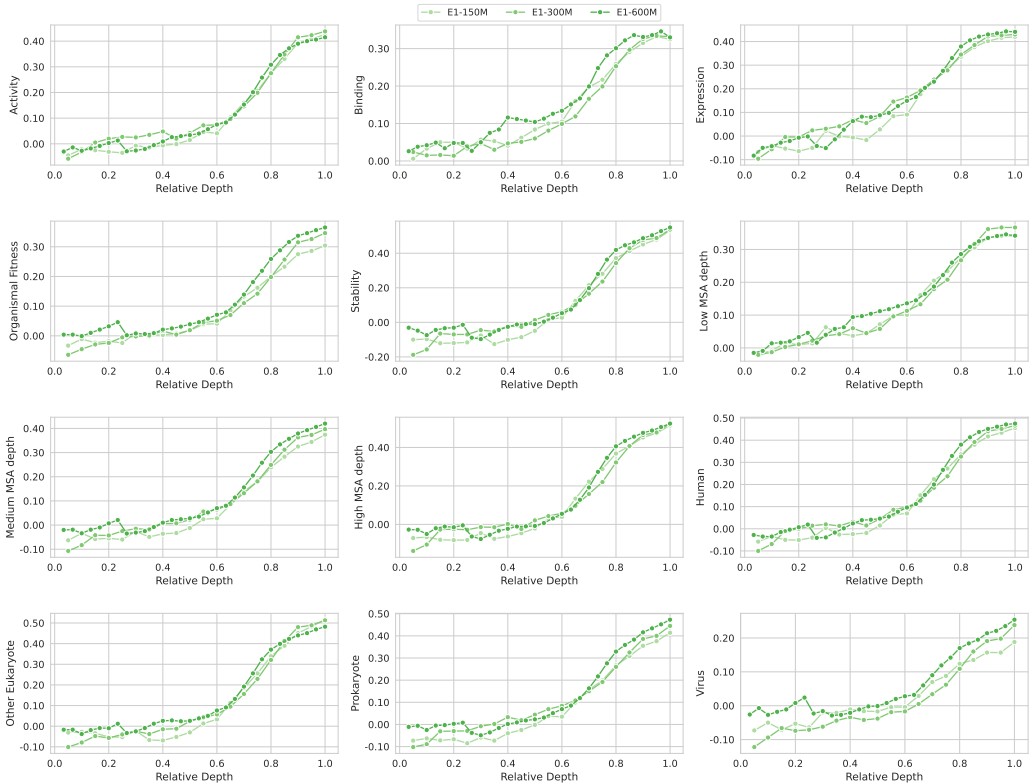

Figure 31: Average Spearman correlation for Profluent-E1 on ProteinGym, computed at each layer and shown separately by phenotype. Relative depth is normalized to $[0, 1]$.

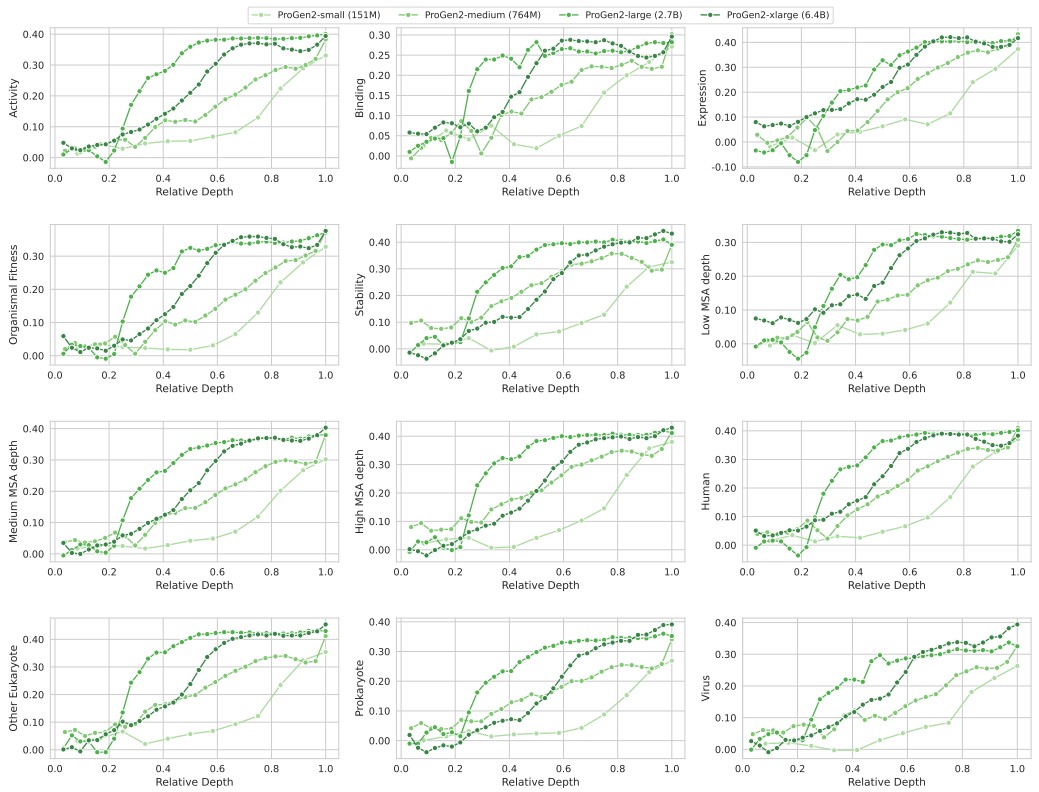

Figure 32: Average Spearman correlation for ProGen2 on ProteinGym, computed at each layer and shown separately by phenotype. Relative depth is normalized to $[0, 1]$.

