# OpenReview forum: "From Words to Amino Acids: Does the Curse of Depth Persist?"
_ICLR.cc/2026/Workshop/LMRL — ICLR 2026 Workshop LMRL Poster_

### Official Review · Reviewer_Dxx7 · 2026-02-16
**Review on Depth Analysis for Protein Language Models**

**Rating:** 5
**Confidence:** 4

**Review:**

This paper explores whether the “curse of depth,” recently identified in Large Language Models, is also present in Protein Language Models (PLMs). It uses different metrics to test the importance of layers across several masked and autoregressive PLMs, in both sequence-only and multimodal models. The paper finds that there seems to be “depth inefficiency,” with later layers progressively refining model predictions.

The authors are exhaustive in their choice of PLMs, and they address a relevant problem (whether some layers in PLMs are less important than others). However, there are some weaknesses:

- The paper seems to be quite incremental compared to the one by Csordas et al. The analysis and conceptual problem are quite similar, except that in this case the study is further restricted to PLMs.
- Did the authors consider measuring performance (e.g., perplexity) after pruning layers? This would be a more straightforward method of assessing the importance of each layer.
- As the authors acknowledge, some important models are missing from the analysis (e.g., AlphaFold, Boltz, but also models like ProstT5).
- Only one downstream task is evaluated in this paper (ProteinGym). The paper would greatly benefit from one or two additional tasks to gain better insight into how the performance-by-layer curve generalizes across different tasks. In particular, for several tasks in biological language models, it has been observed that the best-performing layers are not the final ones, and this is not reflected in the current task.
- Why is ESM3 excluded from the downstream task analysis? Even though it is a multimodal model, it can also function in a sequence-only setting.
- The number of proteins used for these tests seems to be very small (only 40 proteins in total).
- There is no quantitative measure or clear criterion supporting many of the claims. For example, there is no defined threshold for when performance improvements in a layer are considered incremental.
- In my opinion, there is not enough commentary on why ESM3 appears to behave differently from the other models.
- The authors could have explored using a measure of representation similarity across layers to more clearly support the claim that representations do not change substantially in later layers.

Conclusions

The quality of the paper would greatly improve if the authors expanded the number of proteins used in the tests (beyond 40) and introduced more quantitative measures to support their claims (e.g., clearly defining which layers are considered “inefficient” and specifying the ranges in which layers are deemed not to make meaningful contributions).

---

### Official Review · Reviewer_bHV5 · 2026-02-25
**An intersting port of LLM depth effects on PLMs. Important work that needs stronger methods.**

**Rating:** 6
**Confidence:** 5

**Review:**

- Summary of work: This paper investigates whether protein language models waste their deeper layers the same way text LLMs do. They analyze 6 PLM families with a total of 20 model variants spanning masked, autoregressive, and diffusion training objectives. The authors use three measurements: layer-skipping interventions to test how much each layer affects downstream computation, LogitLens readouts to track how the predicted distribution evolves across depth, and layer wise early-exit evaluation on the ProteinGym mutation benchmark. Across all models, they find the same consistent pattern in where early to middle layers do the heavy lifting while later layers mostly refine an already stable prediction, and this effect gets more pronounced as models scale. They conclude that depth inefficiency is a general property of modern PLMs, not just autoregressive text models, motivating future work on more depth-efficient architectures.

- **Strengths:**
* Broad, systematic coverage: 6 PLM families, 20 variants, 3 training objectives. No prior work has done this for protein models at this scale.

- **Weaknesses:**
* Authors never engage with the Pre-LN mechanistic explanation from Sun et al., which identified Pre-Layer Normalization as the root cause of depth inefficiency and showed Post-LN models (BERT) don't exhibit it. Since ESM2/DPLM use Pre-LN, the findings may be a predicted architectural consequence rather than a new discovery
* The skiplayer future intervention means fundamentally different things for autoregressive vs. masked models (temporal compositional reuse vs. cross-position influence). This undermined the unified curse persists narrative. In bidirectional masked or diffusion models, there is no causal chain as all positions attend to all others at every layer,  so the adapted version only tests cross-position propagation. But a late layer in a bidirectional model can show small cross-position effects not because it's redundant, but because early layers already handled cross-position communication through bidirectional attention, and the late layer is doing important local refinement that doesn't need to propagate spatially. The experiment can't distinguish whether late layers are genuinely unnecessary from whether late layers do meaningful local work that the cross-position metric is blind to. So when the paper gets similar-looking heatmaps across autoregressive and masked models and concludes the same phenomenon is occurring, the numerical agreement may be superficial with the same pattern produced by different mechanisms. The ProteinGym early-exit results, which test actual functional utility per layer regardless of objective type, are much stronger evidence for the unified claim, but the skiplayer analysis gets the most emphasis despite being the weakest link in the cross-objective narrative.
* Prior LLM work (Sun at al.) argues Pre-LN dynamics can drive near-identity deep blocks. This paper doesn’t test whether normalization/variance growth explains the PLM patterns, despite studying families where this is plausibly decisive and sustaining its methods in that prior work.
* The multimodal inference discussion is undersupportned. The authors analyze only the sequence pathway even for multimodal models, so they cannot distinguish depth inefficiency in the model from depth inefficiency coming from the language stream, nor test whether cross-modal integration lives late. Totally speculative but ESM3 results might point to a different direction.

**Recommendation:** Work like this is very much needed in the field and while it is sufficient for a workshop I encourage the authors to delve further into the mechanisms behind depth scaling, feature learning and performance. And for future versions I'd also recommend to stregthen the analyses around the skiplayer interventions. Weak accept.

---

### Official Review · Reviewer_Hkpt · 2026-02-26
**Interesting empirical observations**

**Rating:** 8
**Confidence:** 3

**Review:**

This paper explores how deeper language model behaves across layers, observing an interesting phenomenon where later layers mostly augment the output distributions, but do not make as big as a quantitative difference. This suggests that the model might not be using different layers equally, and hence is not using all parameters efficiently.


The experiments are ran in great details, over different architecture, and different parameters. Impact on downstream task are thoroughly examined.

---

### Meta-Review · Area_Chair_U2Kn · 2026-02-28

**Recommendation:** Accept (Poster)
**Confidence:** 4

**Metareview:**

Accept

---

### Decision · Program_Chairs · 2026-03-02

**Decision:**

Accept (Poster)

**Comment:**

Please see the meta-review.